# Exploring the Relationship between Particulate Matter Emission and the Construction Material of Road Surface: Case Study of Highways and Motorways in Poland

**DOI:** 10.3390/ma16031200

**Published:** 2023-01-31

**Authors:** Magdalena Penkała, Wioletta Rogula-Kozłowska, Paweł Ogrodnik, Jan Stefan Bihałowicz, Natalia Iwanicka

**Affiliations:** 1Institute of Technical Sciences and Aviation, State Academy of Applied Sciences in Chełm, 54 Pocztowa Street, 22-100 Chełm, Poland; 2Institute of Safety Engineering, The Main School of Fire Service, Slowackiego Street 52/54, 01-629 Warsaw, Poland; 3Institute of Civil Engineering, Warsaw University of Life Sciences, 166 Nowoursynowska Street, 02-787 Warsaw, Poland

**Keywords:** particulate matter, road dust, abrasion of the road surface, exhaust emissions

## Abstract

Road dust is an important inexhaustible source of particulate matter from traffic and the resuspension of finer particles carried by wind and traffic. The components of this material are of both natural and anthropogenic origin. Sources of particulate pollution are vehicles and road infrastructure. The work aimed to analyze the mass fraction of the finest fractions of road dust (<0.1 mm) collected from highways and expressways with asphalt and concrete surfaces. Sampling points were located in the central and southern parts of Poland. The research material was sieved on a sieve shaker. It has been proven that concrete pavement is less susceptible to abrasion than asphalt pavement. Particles formed under the influence of the erosion of asphalt and concrete belong to the fraction gathering coarser particles than the critical for this research fraction (<0.1 mm). It was found that limiting the area with sound-absorbing screens leads to the accumulation of fine road dust in this place, contrary to the space where are strong air drafts that remove smaller particles from the vicinity of the road. In general, the mass fraction of particles smaller than 100 μm in road dust was from 12.8% to 3.4% for asphalt surfaces and from 12.0% to 6.5% for concrete surfaces.

## 1. Introduction

In recent years, stringent regulations have promoted intensive testing of exhaust emissions, which has led to a marked reduction in particulate emissions from fuel combustion [1,2]. Therefore, the development of vehicle technologies and new low-impact engines, such as electric motors and hybrids, has increased the percentage of non-combustion sources in total emissions from transport sources. Car traffic is the dominant source of particulate matter (PM) emissions in densely populated countries of Western Europe, while in Eastern Europe and some Central European countries, coal and biomass combustion processes are primarily responsible for air dustiness [3]. According to Świetlik and Trojanowska [4], road transport in Poland is responsible for the emission of 73.2 Gg/year of PM, which is 18.6% of total PM emissions [5]. The distribution of particulate matter emitted as part of transport emissions into individual processes that make up the total road emissions is interesting. Combustion of fuels in engines is the source of only 24.8% of PM emissions; the largest share in PM emissions is road surface wear—49.2%, and lower tire wear—24.0%. The use of friction elements in the braking system accounts for only 2.0% of PM emissions [4]. Rexeis and Hausberger [6] have suggested that if the levels of non-exhaust pollutants are not significantly reduced, the percentage of such particles may increase to as much as 90% of particulate traffic pollution. The above list also clearly shows that the processes related to the erosion of the material from which the road surface is constructed and the surface type are of key importance in developing a strategy for reducing PM emissions from communication.

Based on literature reports, it was found that little research has been done on the size distribution of road dust particles so far [7]. Meanwhile, their skillful grouping is extremely important, as it is the first and most important step in processes/programs/scenarios that allow, among others, to reduce PM concentrations in places exposed to its concentration, such as around roads and intersections [8]. Road dust, which has various origins, can vary in size from 0.02 to 10 mm in terms of particle size [9].

The simplest way of grouping road dust is to assign particles of appropriate size ranges to specific sources, which are characterized by lower and upper cut-off diameters [10,11,12]. It is assumed that the post-exhaust emissions mainly contain coarse particles (large particles, larger than 100 µm, but also some particles belonging to PM, including the fraction of coarse dust defined as PM2.5–10; particles with an equivalent aerodynamic diameter in the range of 2.5–10 µm). A negligible mass of road dust from non-exhaust sources consists of particles belonging to the PM2.5 fine dust fraction. Particles of this fraction are particles emitted mainly with car exhaust [13,14,15,16]. Nevertheless, it is now possible to find scientific reports proving that exhaust emissions are also a source of nanoparticles [1,17,18,19]. The size of the emitted particles depends primarily on the physical properties and chemical composition of the material subject to abrasion/erosion, as well as on the type, value and complexity of the forces acting on this material and the temporal and spatial variability of these parameters [20,21].

By classifying PM particles in terms of size only for non-exhaust emission sources, it has been proven that fragments of worn road surface belong to the fraction of PM coarse. At the same time, elements from the abrasion of tires and brake discs supply the fraction of fine and coarse PM [22]. Currently, there are no standards assessing the quality of road dust in terms of quantity, chemical properties and particle size distribution.

One of the basic methods that can be used to determine the share of PM in road dust is to examine its granulometric composition using a simple sieve analysis. The granulometric composition is an indicator of the heterogeneity of dust grain sizes, which can be presented as a graph showing the relative share of grains smaller or larger than a given size d in the total dust mass. This dependence is closely related to the speed of grain falling in calm air. The speed of descent increases significantly with the increase in the diameter of the grains. The residence time of the dust grains in the air depends on the speed of their descent, which is determined by their size and density and is the result of the relationship between the force of aerodynamic drag and the force of gravity [23]. Granulometric analysis is an important parameter by means of which the correctness of many processes is assessed, and its results are an important element of the evaluation and optimization of the applied solutions [24].

PM particles generated during pavement erosion can be in a wide range of sizes, and their composition is determined by the material constituting the building material of the top road layer. Pavements can be broadly classified as concrete or asphalt. The base of most paved roads is the same and consists of an aggregated mixture of various grain sizes, bitumen or cement and modifiers such as fillers and binders. The choice and proportions of the ingredients determine the differences in the exact chemical composition of the mixtures used to produce this road layer [25]. Polymers, epoxy resins, low-carbon steel and sulfur are used as modifiers to increase the hardness of the binding components. To bind the binder with the aggregate, fillers and reinforcing fibers are used, such as glass, fly ash and shredded used tires [1,21,26]. Concrete pavements are a combination of mineral aggregates, sand and cement. There is little literature on the chemical composition of concretes used to make roads and the dust emitted from such surfaces. This is because there are many possibilities for choosing the composition and proportion of materials used to make such a mixture. Consequently, there is no universal molecular formula for road concrete, and therefore no particles are emitted during the erosion of such roads [21]. Since crystalline silica in the form of quartz is the main component of silica dust added to concrete, airborne dust generated during pavement abrasion may have properties that will increase the risk of excessive pollution—e.g., with crystalline silica—of the air and land in the vicinity of the roadway [27].

The aim of the work is to determine the granulometric composition of road dust collected in various regions of Poland from highways/expressways with asphalt and concrete surfaces at three control points: in the space between sound-absorbing screens, space without screens and at road exits. The presented research was aimed at determining the share of PM in road dust depending on the type of pavement and the place of sampling. In addition, the obtained results made it possible to determine what changes occur in the granulometric composition of road dust collected at specific control points. 

Although the scope of the work and the methodology used are not new in the world literature, it should be emphasized that in Poland, such studies have not been conducted so far. Meanwhile, it is precisely in Poland that this problem is important due to the high and still reported dangerously high values of PM concentrations in the atmosphere. In this context, determining the share of PM in road dust and, thus the share of road dust in shaping PM concentrations seems to be extremely important. In addition, determining the relationship between the quality of road construction material and the amount of PM emissions during car traffic may shed additional light on the legitimacy of searching for new materials, additives, or technologies to reduce this emission.

## 2. Materials and Methods

Road dust samples were selected for road sections meeting the highest technical and operational requirements, i.e., highways and expressways located in the central and southern parts of Poland. Roads with two types of the surface were selected (asphalt and concrete). The list of roads is provided in Table 1. The locations of measurement points on the map are shown in Figure 1.

For each section of the highway/expressway, six places were selected where it was possible to collect material for testing. Samples were collected on the left and right sides of the road at three control points:• In the space between sound-absorbing screens (S),• In a space without screens (F) (a typical sampling site is shown in Figure 2),• At road exits (E).

The distances between the control points were about 5 km and were at the approximate height for the left and right sides of the selected road. For the space between the screens, sections were selected that were covered with a barrier on both sides and were located at a point about 2 km from the edge of the screen. The dust was sampled on both sides of the road in order to obtain a representative sample, not affected by wind, not the direction of traffic. The test material was collected manually, each time, from an area of not less than 2 m^2^. In total, samples from 48 points were analyzed. The material for the tests was collected in the summer months due to the reduction of the impact of de-icing agents in road dust.

The study of the granulometric composition of the material was carried out in the laboratory of the National Academy of Applied Sciences in Chełm on a Sieve Shaker mechanical sieve shaker (Figure 3); RADWAG laboratory balance was also used. Using a standard type of sieves, seven material fractions were obtained: 10–2 mm, 2–1 mm, 1–0.5 mm, 0.5–0.25 mm, 0.25–0.1 mm, 0.1–0.063 mm and <0.063 mm.

After performing the granulometric analysis (sieve method) and calculating the percentage content of the mass of grains and particles (with dimensions smaller than the successive diameters di), graining diagrams (graining curves) were prepared in accordance with the PN-EN ISO 14688-1 standard [29]. These plots were plotted on a semi-logarithmic grid, where on the abscissa axis, the diameters of grains and particles were given in a logarithmic scale, and on the ordinate axis in a decimal scale their percentages were given [30]. Graining curves were made by comparing the results for three control points (S, F and E) and for two types of pavement (asphalt/concrete).

Grain size curves are, in fact, empirical distributions of particle mass distribution. To answer the question of whether there are statistically significant differences between the two grain-size curves, the non-parametric Cramér–von Mises test was used [31,32]. Using this test, at each sampling point (measuring points 1–8), the grain size curves at the control points (S, F and E) were compared in pairs. The tests were carried out using the scipy package of the Python programming language [33].

## 3. Results and Discussion

In order to determine the impact of road surface abrasion on the amount of dust pollution accumulating in the vicinity of the right-of-way, the average values of the fractional composition of road dust from three control points (S, F and E) were compiled (based on Table 2). The aggregate results of the two smallest fractions, i.e., 0.1–0.063 mm and <0.063 mm, were used for the analysis. The results for asphalt and concrete roads are shown in Figure 4, Figure 5 and Figure 6.

In the case of routes with an asphalt surface, the largest share of PM in the mass of road dust was obtained for sections 3 and 4 (S8 Wrocław–Sieradz and S17 Lublin–Piaski). These were the results of 12.8% and 11.6%, respectively. At point 1 (A2 Warsaw–Łódź). the share of PM in the total mass of road dust was 9.3%, while the lowest values were obtained for point 2 (S7 Kielce Bypass); it was only 3.4%. It is worth noting that this is the minimum percentage of PM for all analyzed measurement points (both with asphalt and concrete surfaces).

For roads with a concrete surface, the highest share of PM in the mass of road dust (12.0%) was obtained for point 5 (A1 Częstochowa–Katowice). At points 8 and 6 (S8 Sieradz–Wrocław and S8 Warszawa–Piotrków Trybunalski), the share of PM in the mass of road dust was at the level of 9.9% and 9.6%. The smallest share of PM in the mass of road dust was obtained for point 7 (S7 Kraków–Widoma); it was only 6.5%. It is worth noting; however, the results for the concrete pavement were less diverse than for asphalt roads.

Comparing the mass fraction of particles <0.1 mm of material collected from two types of pavement (Figure 6), it was found that the obtained results are very similar. The share of PM in the mass of road dust is slightly higher for the concrete pavement (9.5%), while for the asphalt pavement, the value was 9.3%. Based on the obtained results, it should be concluded that the share of PM in the mass of road dust is not affected by the type of road surface. Particles generated from the abrasion of both asphalt and concrete roads are larger than the analyzed fraction <0.1 mm [22].

Figure 7 compares the share of the finest road dust particles (<0.1 mm). i.e., PM in the mass of road dust for specific control points (S, F and E).

For the area covered with sound-absorbing screens (S). the share of PM in the road dust mass was from 3.9% in point 2 (S7 Kielce bypass) to 19.9% in point 4 (E8 Lublin–Piaski expressway). In the case of the checkpoint where there were no terrain barriers in the vicinity of roads (F), the results oscillated between 2.8% in point 2 (S7 Kielce bypass) to 16.1% in point 3 (E8 Wrocław–Sieradz expressway). On the other hand, at the third control point (E), the share of PM in the road dust mass was from 3.6% in point 2 (S7 Kielce ring road) to 15.6% in point 5 (A1 Częstochowa–Katowice highway) and 8 (S8 Sieradz expressway–Wroclaw). It follows that the largest mass share of the finest particles of road dust occurs in the space covered with a terrain barrier, which is consistent with the results of previous research [34]. It has been proven that noise barriers adjacent to the roadway can inhibit the movement of air off the road, leading to increased concentrations of pollutants on the roads and thus blocking the effective dispersion of the finest particles [35,36]. However, in the case of control points F and E, the maximum values of the share of PM in the mass of road dust are similar and oscillate around 16%. Comparing the measurement points (F-E), a higher share of PM in the mass of road dust was obtained in four points for area F (E8 Wrocław–Sieradz E8 Lublin–Piaski, S8 Warszawa–Piotrków Trybunalski, S7 Kraków–Widoma) and four for area E (A2 Warszawa–Łódź, S7 ring road of Kielce, A1 Częstochowa–Katowice, S8 Sieradz–Wrocław). The vicinity of the control points does not affect the fraction <0.1 mm in the mass of road dust.

In the case of routes with an asphalt surface, the highest share of PM mass in road dust was shown for the checkpoint limited by sound-absorbing screens (S) at the 4th measurement point (E8 Lublin–Piaski expressway) with a value of 19.9%, while the smallest for the checkpoint not limited by a barrier off-road (F) at the second measurement point (S7 Kielce bypass) with a value of 2.8%. For roads with a concrete surface, the largest share of PM in road dust was obtained for the control point located near the exit (E) from the highway/expressway at the 5th (A1 Częstochowa–Katowice highway) and 8th (S8 Sieradz–Wrocław expressway) measuring points; it was a share of 15.6%. The lowest values were shown for the control point covered with sound-absorbing screens (S) at the second measurement point (S7 bypass of Kielce), with a value of 4.7%. In the case of the area covered with sound-absorbing screens (S), it was noticed that the share of PM in road dust collected on sections with concrete pavement is lower but much less diverse than in the case of dust from asphalt pavements. This is due to the use of high-strength concrete in the construction of highways and expressways [37]. In order to obtain concrete mixes resistant to abrasion, appropriate concrete compressive strength, number and type of air pores, water-cement index, and type and properties of the aggregate used (mainly its quantity and hardness) are required [38]. Also required are dosage and composition mixtures [39], the use of supplementary cementing materials such as fly ash [38], silica fume [27], the addition of polymer and resin fibers, steel filings, ground silicon carbide, corundum, porcelain cullet and lead or copper slags [40].

In the case of points located in the receptors of the so-called free space (F), the share of PM in the mass of road dust for concrete pavement is much less differentiated than in the case of asphalt pavement. Moreover, when comparing the control points of screens-free space (S-F) for the concrete pavement, the share of PM in road dust is lower in each case for screens (S) than for free space (F). On the other hand, for the asphalt surface, the reverse trend is observed S > F (in three out of four cases). Therefore, it should be stated that in the area of free space (F), a significant role is played by air ducts that blow away lighter particles from the vicinity of the right-of-way. The settling speed of dust particles increases significantly with the increase in grain diameter [23]. This may mean that the process of concrete pavement erosion is a source of particles with a larger diameter than those generated in the abrasion of asphalt pavements.

For points located in the vicinity of exits (E) from a highway/expressway, the share of PM in the mass of road dust for a concrete pavement is much higher than in the case of an aslt pavement. This is due to the mechanical properties of the concrete surface,. i.e., good adhesion and a shorter braking distance (also on a wet surface) compared to the asphalt surface [41]. The tire-to-road adhesion coefficient μ for dry concrete is 0.8–1.0, while for dry asphalt, it is 0.7–0.8 [42]. It follows that the concrete surface is rougher than its asphalt counterparts and can generate an increased amount of particles generated during vehicle braking [43]. In addition, there are expansion joints on concrete roads, which cause discontinuity of the surface, which is conducive to the process of tire and pavement wear [44].

The granulometric composition of the dust collected at each measurement point was presented in the form of particle size distribution curves [30] and summarized for three control points in Figure 8 for the asphalt pavement and Figure 9 for the concrete pavement.

Based on the grading curves, it was found that at points 1 (Warsaw–Łódź A2 highway) and 3 (E8 Wrocław–Sieradz expressway), the curves of the control point covered with sound-absorbing screens (S) differ from the others. In addition, the grain size curves for the checkpoint located near the exit at points 4 (E8 Lublin–Piaski expressway) and 5 (A1 Częstochowa–Katowice highway) are shifted relative to the others, but to a lesser extent than for the above-mentioned points at the checkpoint covered with screens. In addition, the differences between the graining curves at points located near the concrete pavement are smaller. The most similar distributions were obtained for point 2 (S7 Kielce Bypass—asphalt surface) and point 7 (S7 Kraków–Widoma expressway-concrete surface).

The granulometric composition of the dust collected at each measurement point is presented in the form of grain size curves [30] and summarized for three control points (S, F and E) in Figure 10, Figure 11 and Figure 12.

The analysis of the obtained results allows us to suspect that for points located in the vicinity of screens (S), there is no dependence of the grain distribution on the type of surface, while for points located in free space (F) and in the vicinity of exits (E) it seems that grain size curves differ between types of surfaces. To prove this, the Cramér–von Mises test was carried out for two cumulative distribution–grain size curves. The p-values of the test for the measurement points (1–8) comparing the control points (S, F and E) in pairs are presented in Table 3. The *p*-values obtained in this test were always at least 0.72, which means that there are no grounds to reject the hypothesis that the grain size curves at the control points do not differ. It can be observed that the *p*-values for the concrete pavement always exceed 0.9, and in 5 out of 12 comparisons, they exceed 0.995, while for the comparisons of the control point of the screens S at measurement points 1 (A2 Warsaw–Łódź) and 4 (E8 Lublin–Piaski), one can see *p*-values ≤ 0.81.

To check the influence of the type of pavement at a given control point, the Cramér–von Mises test was performed again and its results are presented in Table 4, Table 5 and Table 6. In this case, high *p*-values were also observed (always *p* ≥ 0.65), and for the checkpoint located near the exits (E), even *p* ≥ 0.72.

The high *p*-values in both Table 3 and Table 4, Table 5 and Table 6 may be related to the nature of the method by which the dust was analyzed, i.e., with the number of sieves used in the granulometric analysis. For a larger number of points, the sensitivity of the test increases automatically, as the currently observed differences between the distributions will result in lower values of the *p* parameter because the statistic used in this test for two samples of equal size is inversely proportional to the square of the number of measurement points [32] *p*-value check.

It was noticed that the most similar distributions occur for the concrete pavement for the S–F graining curves, and the least for the asphalt pavement for the S–E relation. This may be due to the fact that concrete pavement is less susceptible to abrasion than asphalt pavement because the contaminants that accumulate on it have similar grading curves regardless of the control point (S and F), in which the air ducts play an important role [37]. On the other hand, for asphalt pavements, the greatest differences were observed for the S–E relation, i.e., at points where factors such as increased vehicle braking (for the exit area) and the impact of airways (smaller for the area of screens and greater for the area of exits) occur. Asphalt surfaces are more susceptible to mechanical abrasion caused by vehicle braking. Particles generated under the influence of this road maneuver belong to the fraction gathering coarser particles than the critical fraction of particles <0.1mm for the purpose of the research [23].

## 4. Conclusions

Test results proved that the concrete surface is less susceptible to abrasion than the asphalt one. In addition, the particles generated by the erosion of both asphalt and concrete roads are likely to be in larger size ranges than the analyzed fraction <0.1 mm. It has also been shown that limiting the area with sound-absorbing screens affects the accumulation of finer dust particles in this place, in contrast to the space where strong air drafts remove smaller particles from the vicinity of the roadway. It is worth noting that the collection of road dust took place in the summer. Therefore, the material collected at that time was not enriched with winter road maintenance agents (e.g., road salt) and was the most reliable in terms of testing the material for surface abrasion. In addition, road dust was collected on sections of basically continuous traffic, which made it possible to significantly eliminate from the mass of collected dust those particles that arise from the abrasion of brake elements, bodywork, or tires (mainly during braking).

Comparing the results from three control points, it was found that the share of PM in the mass of road dust for the point covered with sound-absorbing screens (S) ranged from 3.9% to 19.9% for the area not limited by a terrain barrier (F) from 2.8% to 16.1%, while for a point located in the vicinity of an exit from the highway/expressway (E) from 3.6% to 15.6%. The shares of PM in the mass of road dust were relatively small, which does not mean, however, that they did not translate into high concentrations of PM in the air near the right-of-way. It is clear that road dust is a source of PM, but the research should be extended to the separation and testing of fractions of particles smaller than 10 μm and smaller than 2.5 μm in the PM of road origin. Based on the data obtained during this study, it is recommended to continue the research about PM emitted during the abrasion of road surfaces, both asphalt and concrete.

## Figures and Tables

**Figure 1 materials-16-01200-f001:**
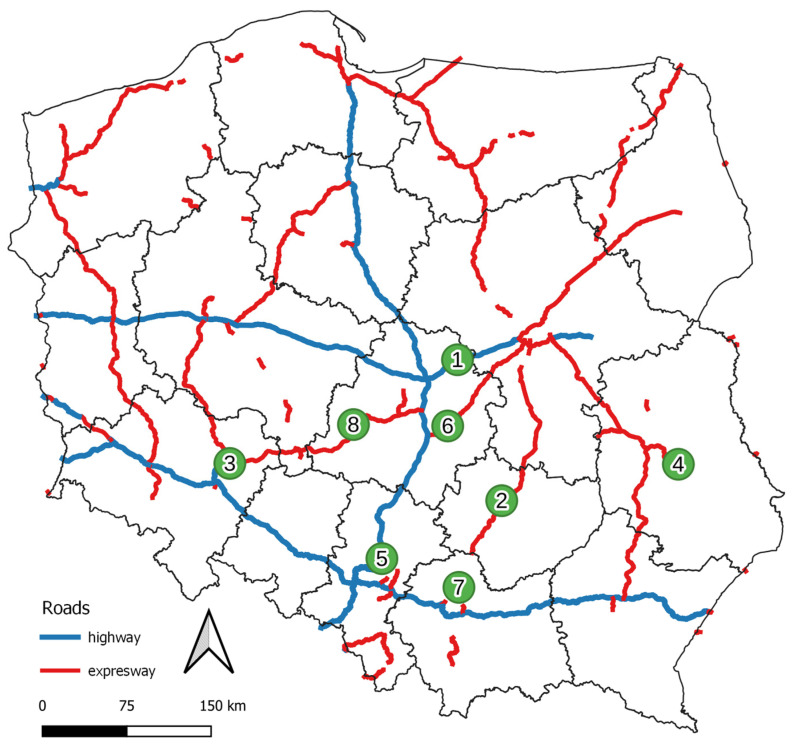
Map of Poland with the markings of roads selected for research background map from [28].

**Figure 2 materials-16-01200-f002:**
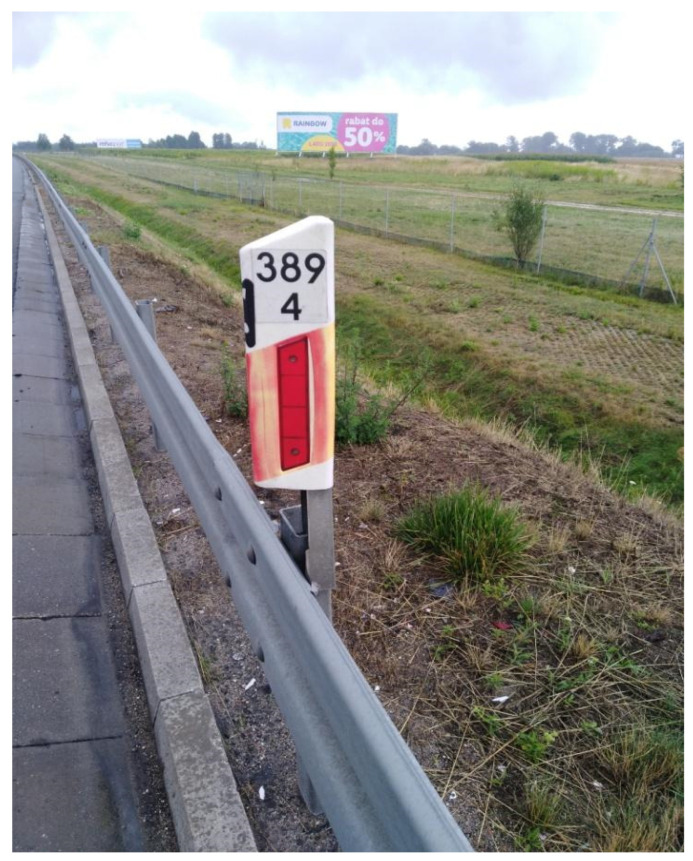
Measuring point at Łódź–Warszawa–free space (F).

**Figure 3 materials-16-01200-f003:**
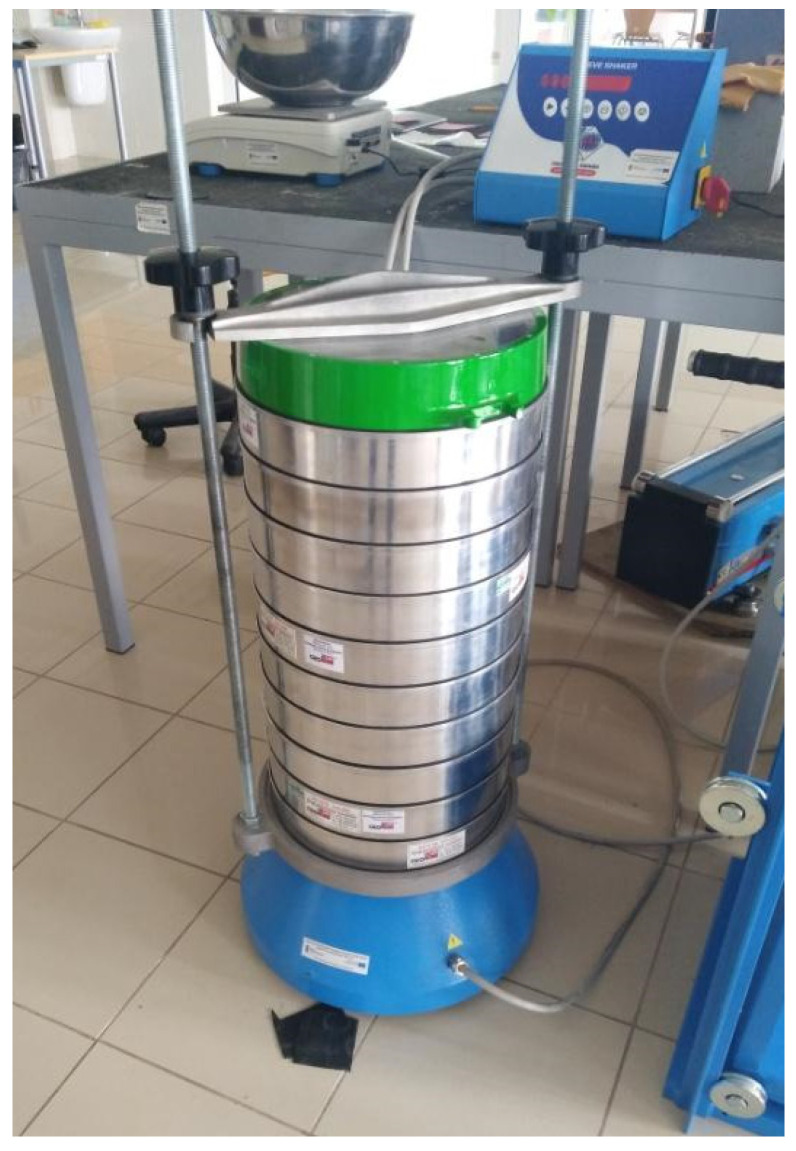
The Sieve Shaker mechanical sieve shaker.

**Figure 4 materials-16-01200-f004:**
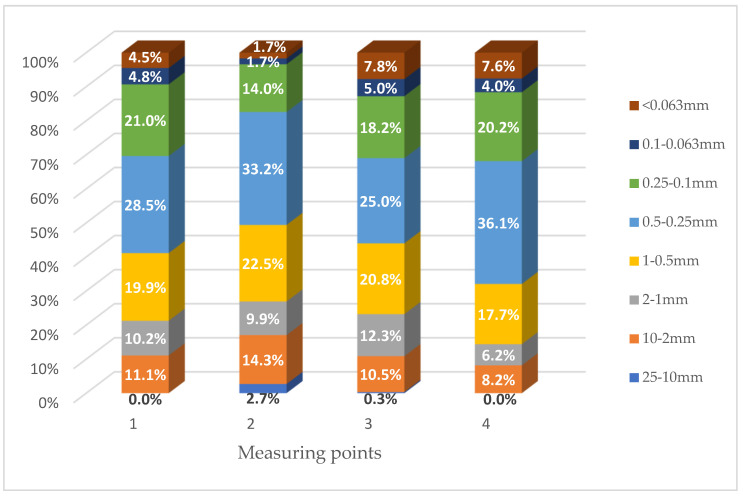
Share of size fractions in the mass of the tested road dust for measurement points with an asphalt surface.

**Figure 5 materials-16-01200-f005:**
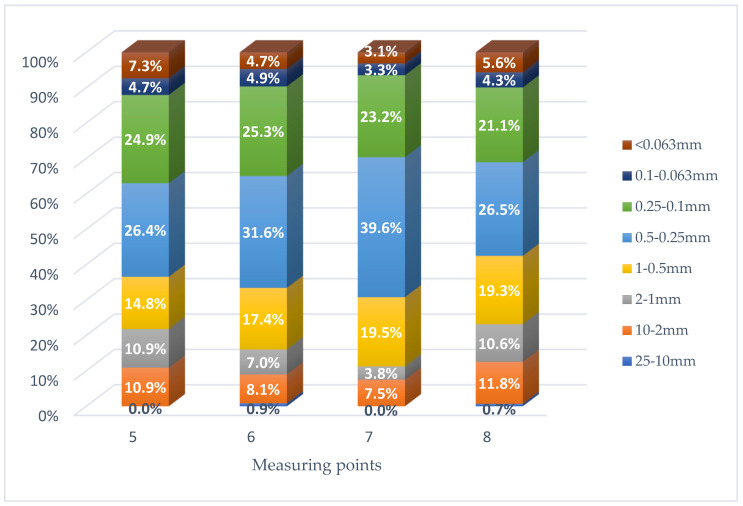
Share of size fractions in the mass of the tested road dust for measurement points with a concrete surface.

**Figure 6 materials-16-01200-f006:**
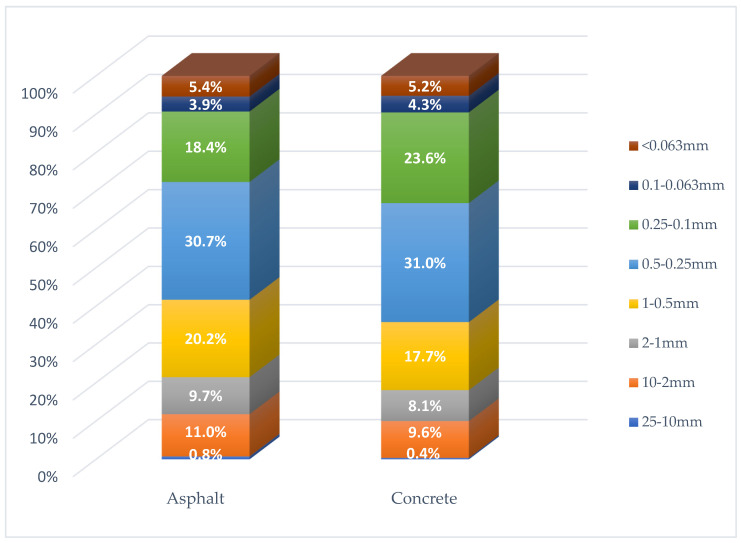
Share of size fractions in the mass of the tested road dust for measurement points with asphalt and concrete surfaces.

**Figure 7 materials-16-01200-f007:**
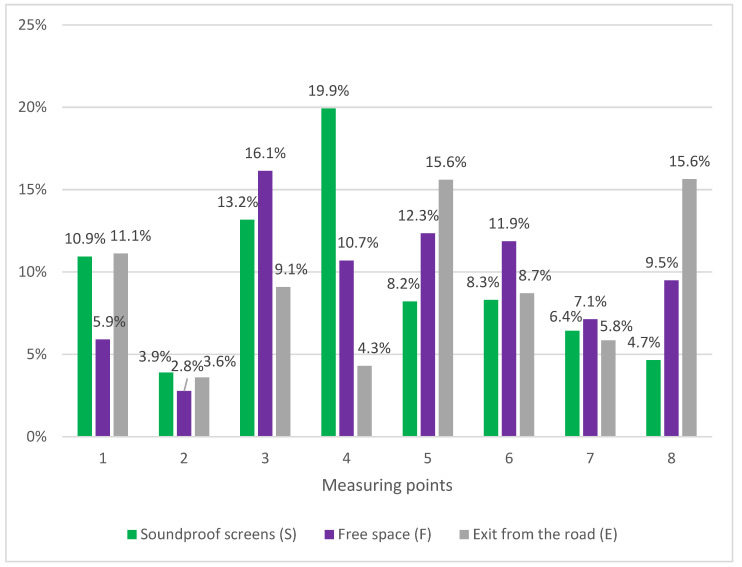
The share of the finest road dust particles (<0.1 mm) in the PM mass for selected control points (S, F and E).

**Figure 8 materials-16-01200-f008:**
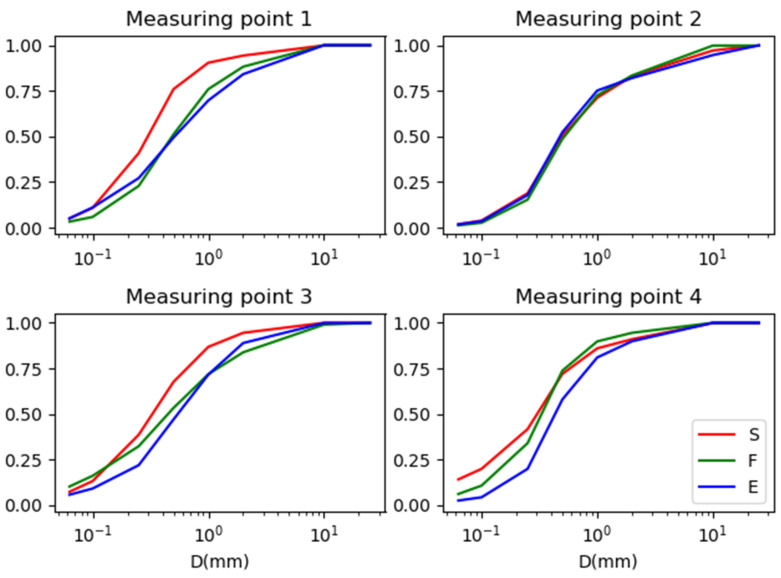
Graining curves for three control points (S, F and E) for measurement points with an asphalt surface. S—area covered with sound-absorbing screens; F—area without soundproofing screens (or other terrain barriers); E—exit from the highway/expressway.

**Figure 9 materials-16-01200-f009:**
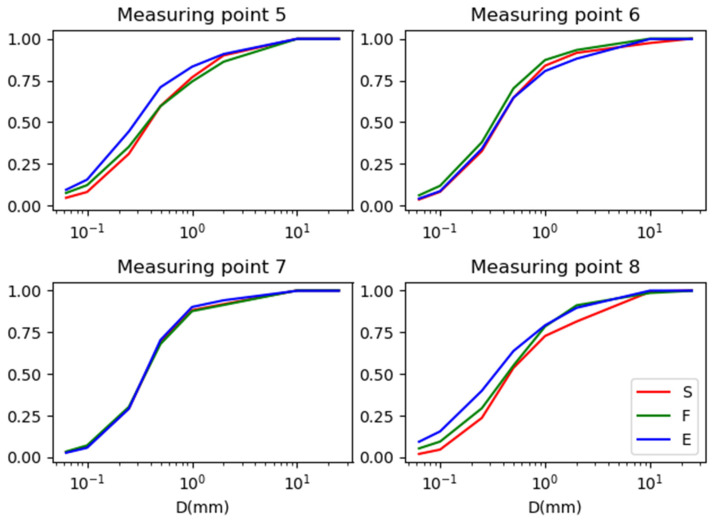
Graining curves for three control points (S, F and E) for measurement points with concrete pavement. S—area covered with sound-absorbing screens; F—area without soundproofing screens (or other terrain barriers); E—exit from the highway/expressway.

**Figure 10 materials-16-01200-f010:**
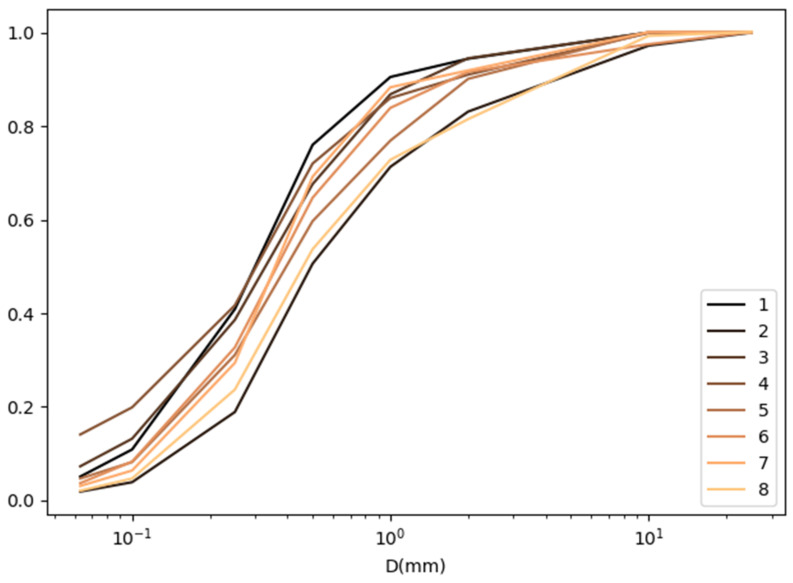
List of graining curves at the control point Screens (S).

**Figure 11 materials-16-01200-f011:**
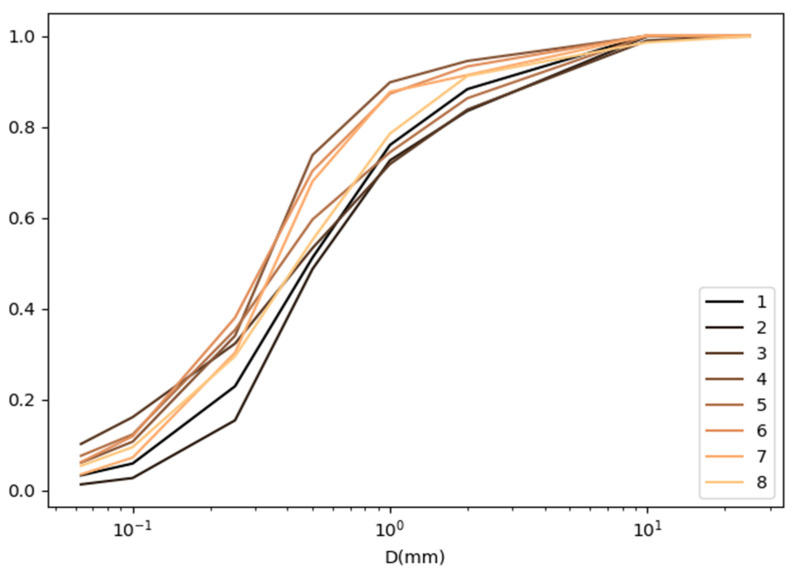
List of graining curves at the Free Space control point (F).

**Figure 12 materials-16-01200-f012:**
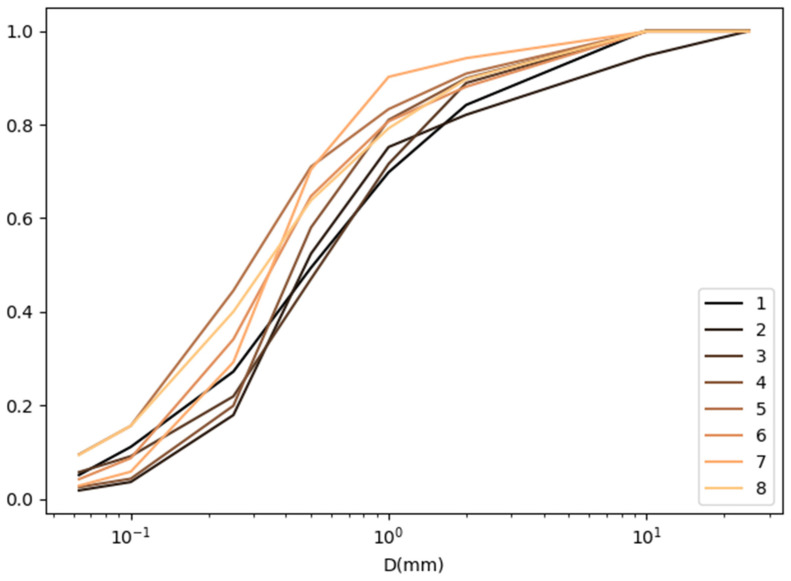
List of grain curves at the exit from the road control point (E).

**Table 1 materials-16-01200-t001:** Roads selected for research.

Measuring Point	Road No	Selected Roads	Section of the Road	Surface Type
1	A2	Warszawa–Łódź	The Łowicz junction–Skierniewice junction	asphalt
2	S7	The bypass of Kielce	The West Kielce–Kielce–Jaworznia junction	asphalt
3	S8	Wrocław–Sieradz	The Wrocław–Psie Pole junction–Oleśnica–West junction	asphalt
4	S17	Lublin–Piaski	Piaski bypass	asphalt
5	A1	Częstochowa–Katowice	The Woźniki junction–Pyrzowice junction	concrete
6	S8	Warszawa–Piotrków Trybunalski	The Wolbórz junction– South Tomaszów Mazowiecki junction	concrete
7	S7	Kraków–Widoma	Wesoła–Widoma–Kraków	concrete
8	S8	Sieradz–Wrocław	The Złoczew junction–South Sieradz juncton	concrete

**Table 2 materials-16-01200-t002:** The fractional composition of road dust at eight measurement points is divided into three control points.

Surface Type	Asphalt				Concrete			
Measuring point	1	2	3	4	5	6	7	8
Road No	A2	S7	S8	S17	A1	S8	S7	S8
Selected roads	Warszawa–Łódź	Obwodnica Kielc	Wrocław–Sieradz	Lublin–Piaski	Częstochowa–Katowice	Warszawa–Piotrków Trybunalski	Kraków–Widoma	Sieradz–Wrocław
Fraction [mm]	**Soundproof screens (S)**							
25–10	0.0%	2.8%	0.0%	0.0%	0.0%	2.6%	0.0%	0.8%
10–2	5.6%	14.1%	5.5%	8.9%	9.9%	5.9%	8.2%	17.8%
2–1	3.9%	11.8%	7.7%	5.0%	13.2%	7.7%	3.6%	8.7%
1–0.5	14.5%	20.7%	19.2%	14.0%	17.2%	19.2%	19.1%	19.1%
0.5–0.25	35.1%	31.7%	29.0%	30.3%	28.6%	32.0%	39.8%	30.0%
0.25–0.1	30.0%	15.0%	25.4%	21.8%	22.9%	24.4%	23.0%	19.0%
0.1–0.063	5.8%	2.0%	5.9%	5.8%	3.5%	4.6%	3.3%	2.6%
<0.063	5.1%	1.9%	7.3%	14.1%	4.7%	3.7%	3.1%	2.1%
<0.1	10.9%	3.9%	13.2%	19.9%	8.2%	8.3%	6.4%	4.7%
Fraction [mm]	**Free space (F)**							
25–10	0.0%	0.0%	1.0%	0.0%	0.0%	0.0%	0.0%	1.3%
10–2	11.8%	16.4%	15.2%	5.5%	13.7%	6.8%	8.7%	7.4%
2–1	12.3%	10.9%	12.0%	4.7%	11.9%	6.0%	3.7%	12.7%
1–0.5	24.8%	23.9%	18.5%	16.0%	14.8%	17.0%	19.7%	23.4%
0.5-0.25	28.3%	33.3%	21.0%	39.8%	24.3%	32.3%	37.7%	25.6%
0.25–0.1	17.0%	12.7%	16.2%	23.3%	23.0%	26.1%	23.1%	20.0%
0.1–0.063	2.6%	1.4%	5.9%	4.6%	4.7%	5.7%	3.7%	4.1%
<0.063	3.3%	1.3%	10.2%	6.1%	7.6%	6.2%	3.5%	5.4%
<0.1	5.9%	2.8%	16.1%	10.7%	12.3%	11.9%	7.1%	9.5%
Fraction [mm]	**Exit from the road (E)**							
25–10	0.0%	5.3%	0.0%	0.0%	0.0%	0.0%	0.0%	0.0%
10–2	15.9%	12.6%	11.0%	10.1%	9.1%	11.8%	5.7%	10.3%
2–1	14.4%	6.9%	17.3%	8.9%	7.6%	7.4%	4.0%	10.5%
1–0.5	20.4%	22.8%	24.6%	23.0%	12.3%	16.0%	19.8%	15.4%
0.5–0.25	22.2%	34.5%	25.1%	38.1%	26.5%	30.6%	41.2%	23.8%
0.25–0.1	16.1%	14.3%	12.8%	15.6%	28.9%	25.4%	23.4%	24.4%
0.1–0.063	6.0%	1.8%	3.4%	1.8%	6.1%	4.5%	3.0%	6.2%
<0.063	5.1%	1.8%	5.7%	2.5%	9.5%	4.2%	2.8%	9.4%
<0.1	11.1%	3.6%	9.1%	4.3%	15.6%	8.7%	5.8%	15.6%

**Table 3 materials-16-01200-t003:** The *p*-values of the Cramér–von Mises test at eight measurement points.

Surface Type	Measuring Points	S–F	S–E	F–E
Asphalt	1	0.81	0.72	1.00
	2	0.98	1.00	0.98
	3	0.98	0.91	0.91
	4	0.81	0.81	0.98
Concrete	5	1.00	0.98	1.00
	6	1.00	1.00	0.91
	7	1.00	0.91	0.91
	8	0.98	0.91	0.91

S—area covered with sound-absorbing screens; F—area without soundproofing screens (or other terrain barriers); E—exit from the highway/expressway.

**Table 4 materials-16-01200-t004:** The *p*-values of the Cramér–von Mises test to determine whether there is an effect of the measurement point and type of pavement on the grading curves at the screen control point (S).

		Asphalt				Concrete			
		1	2	3	4	5	6	7	8
Asphalt	1	1.00	0.81	1.00	0.81	0.98	0.98	0.91	0.72
	2		1.00	0.91	0.65	0.98	0.98	0.81	1.00
	3			1.00	0.81	1.00	0.98	0.81	0.81
	4				1.00	0.81	0.91	0.81	0.81
Concrete	5					1.00	0.98	0.91	0.98
	6						1.00	1.00	0.98
	7							1.00	0.98
	8								1.00

**Table 5 materials-16-01200-t005:** The *p*-values of the Cramér–von Mises test to determine whether there is an effect of the measurement point and type of pavement on the grading curves at the screen control point (F).

		Asphalt				Concrete			
		1	2	3	4	5	6	7	8
Asphalt	1	1.00	0.81	0.81	0.72	0.81	0.98	1.00	0.91
	2		1.00	0.81	0.65	0.81	0.72	0.72	0.98
	3			1.00	0.91	1.00	0.81	0.72	0.98
	4				1.00	0.98	0.91	0.91	0.91
Concrete	5					1.00	0.81	0.72	0.91
	6						1.00	1.00	0.91
	7							1.00	0.91
	8								1.00

**Table 6 materials-16-01200-t006:** The *p*-values of the Cramér–von Mises test to determine whether there is an effect of the measurement point and type of pavement on the grading curves at the screen control point (E).

		Asphalt				Concrete			
		1	2	3	4	5	6	7	8
Asphalt	1	1.00	0.81	0.91	0.81	0.81	0.91	0.72	0.91
	2		1.00	0.91	1.00	0.91	0.91	0.91	0.98
	3			1.00	0.81	0.81	1.00	0.98	0.81
	4				1.00	0.98	0.91	0.81	0.98
Concrete	5					1.00	0.81	0.81	1.00
	6						1.00	0.98	0.81
	7							1.00	0.72
	8								1.00

## Data Availability

Data are available for request from corresponding author.

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
