# Peer review of "Exploring the Relationship between Particulate Matter Emission and the Construction Material of Road Surface: Case Study of Highways and Motorways in Poland"

_materials, 2023, doi:10.3390/ma16031200_

Round 1

Reviewer 1 Report

1.      Revise the title and remove the words “research focus”.

2.      Abstract must be within the journal limits (bout 200 words) and must cover without headings: Background, problem statement, Methods, Results and Conclusion.

3.      Define the abbreviation initially.

4.      Show through figure the sampling location from the road with a few samples.

5.      Revise the Table 1 description.

6.      Figure numbering must be consistent. Figure 6-9, figure 10-13

7.      Existing plots (6-9 and 10-13) are blurt.

8.      Concise the conclusion and keep it in bullet form.

Author Response

Dear Reviewer 1, many thanks for the remarks, it pointed out shortcomings of our article. Please find the detailed response to your remarks below.

1.  Revise the title and remove the words “research focus”.

We changed the title according to your remark.

2.  Abstract must be within the journal limits (bout 200 words) and must cover without headings: Background, problem statement, Methods, Results and Conclusion.

Indeed, the abstract was much too long, so we cutted it.

3.  Define the abbreviation initially.

We corrected the abbreviations according to journal policy https://www.mdpi.com/journal/materials/instructions

4.  Show through figure the sampling location from the road with a few samples.

We added a figure with a photo of typical sampling points.

5.  Revise the Table 1 description.

We revised the description

6.  Figure numbering must be consistent. Figure 6-9, figure 10-13

We changed the numbering.

7.  Existing plots (6-9 and 10-13) are blurt.

We do not know why it happened. We replaced figures.

8.  Concise the conclusion and keep it in bullet form.

We do not agree that conclusions should be in a bullet form and we did not change it. If the editor would like to change it according to you remark we can do it.

Reviewer 2 Report

1) The aim of the work is to determine the granulometric composition of road dust from highways/expressways. The authors state that the size of the emitted particles depends, among other things, on the type, value and complexity of the forces acting on this material and the thermal and spatial variability of these parameters. It is a very broad range of influences. Were certain measurement conditions observed, such as the weight of vehicles or the composition of the given highway, the age of the road?
2) Is it possible to add the table to the map of the names of the monitored roads with the corresponding number?
3) It is a nice and detailed study for practice in the field of road construction. However, the article lacks a scientific recommendation for the future.

Author Response

Thank you for your revision, according to your remarks we added some sentences in the main body, please find the detailed response below.

1) The aim of the work is to determine the granulometric composition of road dust from highways/expressways. The authors state that the size of the emitted particles depends, among other things, on the type, value and complexity of the forces acting on this material and the thermal and spatial variability of these parameters. It is a very broad range of influences. Were certain measurement conditions observed, such as the weight of vehicles or the composition of the given highway, the age of the road?

We are aware that size distribution can be dependent on many parameters. We believe that the traffic that influences the size distribution is rather dependent on the category of road than location, i.e., traffic is similar on motorways. The age of the road is not an important factor for it since all the roads that are in the study are younger than 15 years.

2) Is it possible to add the table to the map of the names of the monitored roads with the corresponding number?

Yes, it was possible, we added it.

3) It is a nice and detailed study for practice in the field of road construction. However, the article lacks a scientific recommendation for the future.

We added sentences in the conclusion section.

Reviewer 3 Report

This is an experimental work that measures particulate matter (PM) generated from different parts of highways. Overall, it is well presented and draws meaningful conclusions supported by the measurement. In addition, original data contributions like this are often valuable for practical engineering problems and the seek of improvements.

I have only some minor comments, mostly on presentation and cosmetic issues.

  • As an experimental work, the reviewer suggest adding one more figure composed of photos of the sampling site and the lab setup.

  • As a map, Fig. 1 should at least include the scale bar. If it does not clutter the figure, some major road network can be shown as well. Lastly, perhaps it is better to put the road names with labels 1-8 as part of the caption instead of the main text---this makes the figure and its caption more coherent and self-contained.

  • Fig. 2 and others: decimal points should be used in place of “,”

  • Fig. 6 is labeled as Fig. 6-9 (?). Its resolution should be increased, and the number after “Measuring point” has incorrect size.

  • Table 3 and 4 have a mixed use of decimal point and “,”.

Author Response

Dear reviewer, thank you for the remarks on our article. We appreciate the time you spend on reviewing our article. Please find the detailed answers below.

As an experimental work, the reviewer suggest adding one more figure composed of photos of the sampling site and the lab setup.

We added a photo of a typical sampling point.

As a map, Fig. 1 should at least include the scale bar. If it does not clutter the figure, some major road network can be shown as well. Lastly, perhaps it is better to put the road names with labels 1-8 as part of the caption instead of the main text---this makes the figure and its caption more coherent and self-contained.

We created a new map according to your remarks.

Fig. 2 and others: decimal points should be used in place of “,”

We revised the decimal separator in all places.

Fig. 6 is labeled as Fig. 6-9 (?). Its resolution should be increased, and the number after “Measuring point” has incorrect size.

We don't know why the quality of figures dropped, we provided new figures.

Table 3 and 4 have a mixed use of decimal point and “,”.

We revised the decimal separator in all places.